# Conceptual Design of a Pilot Process for Manufacturing Aluminum-Based Intermetallic Compound Powders

**Melchor Salazar** [1,*] **and Flavio-Américo Lagos** [2]

1   Centro de Innovación e Integración de Tecnologías Avanzadas, Unidad Papantla Veracruz, Veracruz 93400, Mexico
2   Instituto Mexicano del Petróleo, Mexico City 07730, Mexico; flagos@imp.mx
*   Correspondence: msalazarm@ipn.mx; Tel.: +52-55-5729-6000

**Abstract:** The FeAl intermetallic compound is of great interest for industry due to its low density, low cost and high mechanical and corrosion resistance, so it can replace stainless steels and nickel-based alloys for some applications. In previous publications, the concept (principle) test for a novel FeAl powder manufacturing process has been shown. It consists mainly of the following stages: (a) metallic strip manufacture through rapid solidification, (b) water vapor exposure of these metallic strips for their disintegration and powder generation and (c) powder drying. Experimental tests were performed for 2 g of the FeAl intermetallic compound. However, this process can be extended to manufacture any other intermetallic compound containing aluminum, such as TiAl, NiAl, CoAl or any other that can be obtained from every element that can combine with aluminum, if the aluminum content is between 55 and 60 at.%. Nowadays, this process is at technology readiness level (TRL) 3. Therefore, in this paper, a process equipment up-scaling configuration for producing up to 15 kg powder is proposed. This manufacturing process is an industrial alternative to those commonly used to produce powders of this type of intermetallic compounds, such as mechanical alloying (MA). Moreover, several alternatives for employing renewable energy sources are given, making it even more environmentally sustainable.

**Keywords:** intermetallic compound; FeAl; rapid solidification; up-scaling; solar energy

## 1. Introduction

FeAl intermetallic compounds have attracted great interest [1–7] because of their outstanding corrosion resistance, wear resistance, low density and low cost of their raw materials, which make them an attractive alternative to more expensive materials, such as stainless steels for certain applications [8–10]. However, they show low ductility at room temperature. Therefore, different processes have been used for their production to improve their ductility [11–14], such as MA [15,16]. In former works published by the authors [17,18], process conceptualization tests for the production process have been undertaken with 2 g of material. The introduced novel process offers several benefits, such as energy savings and sustainability [18]. The novel production process takes advantage of the mechanisms affecting FeAl ductility. Currently, this technology is at TRL 3. Therefore, in this paper, a concept of process scaling for a capacity of up to 15 kg is proposed. We present a set-up of the necessary equipment and a description. Moreover, solar energy use is proposed in a considerable part of the process as a proposal for further sustainability for this process.

## 2. Equipment and Process Description

As mentioned in previous publications [17,18], the novel FeAl intermetallic compound powder production process consists of three main steps:

(a)   High-purity iron and aluminum scrap melting;

(b)  Metallic ribbons formation through rapid FeAl solidification;
(c)  Disintegration of FeAl ribbons to obtain powder.

In previous papers, we showed the principle on which this production process is based. It is proposed on a gram-scale, and in this article, a pilot-scale process for FeAl powder production up to 15 kg is proposed.

### 2.1. General Equipment Description

The proposed equipment configuration for FeAl powder production consists of a system composed mainly of four modules, as shown in Figure 1, which are as follows:

I.    Induction furnace;
II.   Melt spinner (rapid solidification);
III.  Steam chamber;
IV.   Powder receiver and conditioning.

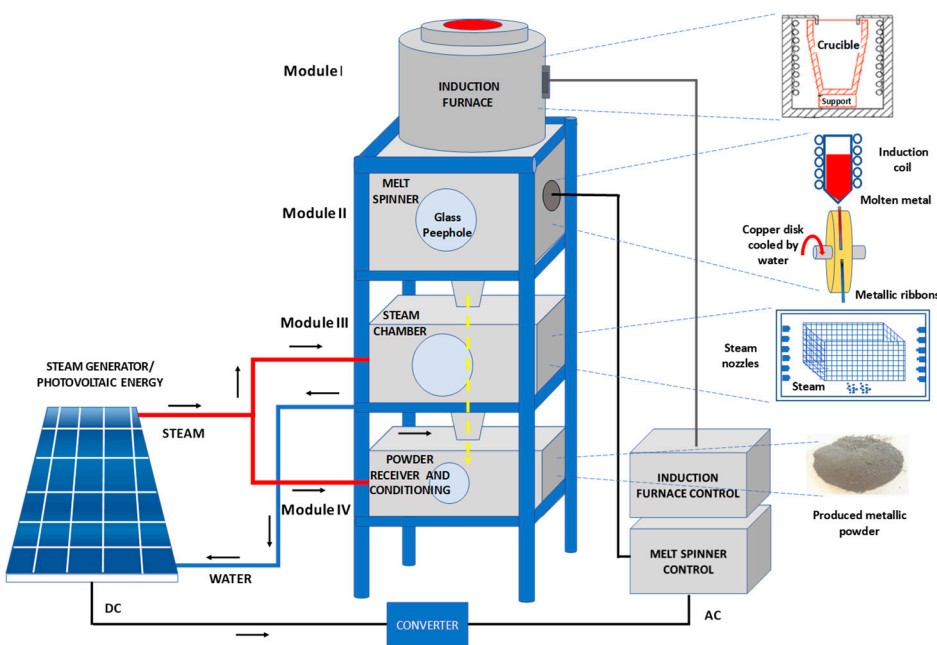

**Figure 1.** Equipment set-up to produce aluminum-based intermetallic powders (55–60%at).

### 2.1.1. Induction Furnace Modulus

This module consists of an induction furnace with a capacity of 15 kg. Energy for the induction furnace can be supplied by:

(a)  Direct connection from an electrical network;
(b)  Photovoltaic (PV) energy supplied by solar cells [19,20];
(c)  Thermoelectric or thermosolar (T) energy.

The use of solar energy as supply energy in the different stages of the process is proposed. Industrial solar systems have been used in the repowering of steam power plants [21–25]. The operation of a thermosolar plant is like that of a thermoelectric central, but instead of using gas or coal, it employs sun energy. Solar energy is concentrated by reflecting surfaces in a receiving device, which reaches temperatures up to 1000 °C. This energy heats a fluid and generates steam, which moves a turbine and produces electricity.

Another alternative for thermosolar energy is to concentrate sun energy using mirrors in the form of a parabolic/concave dish to concentrate the solar rays into a receiver, a Stirling motor [26], placed on the focal point of the parabola. The mirror counts with a biaxial tracking system and can track the sun with a high precision grade to reach high efficiency levels. Heat increases the air temperature up to 650 °C, activating the Stirling

motor. This type of solar system can be considered as an auxiliary for electric energy feed to the induction furnace.

Another way to obtain the metal molten is using the system proposed by Ozcan [27]. Solar systems for aluminum metal melting use feasible concentrated solar energy systems. They use solar tracking in two dimensions with a central receiver and high-temperature furnaces.

PV systems can be used to generate electrical current to feed the induction furnace and for moving the melt spinner disk and other auxiliary equipment used for the operation.

The furnace is fed with aluminum and iron scraps of high purity to produce FeAl intermetallic compound powder (55–60 Al at.%). Once the aluminum and iron are molten, the FeAl intermetallic is formed, as demonstrated in previously published works [17,18]. The molten material flow is transferred (conducted) to the melt spinner module (rapid solidification process), which is controlled through a smaller crucible inside. The rapid solidification process has been used successfully to produce intermetallic compounds, among other materials [28–30].

### 2.1.2. Melt Spinning Modulus

The FeAl molten metal, mentioned in the previous section, is fed into a smaller crucible placed inside of the melt spinner module to regulate its flow. The smaller crucible is heated by electrical resistance to avoid melt solidification.

The melt spinner module also has a disk of a conducting metallic material, such as copper, which is cooled by water. It rotates at a velocity between 5000 and 8000 rpm. The molten material is fed into the disk, which rotates at a controlled speed.

As a result of the rapid solidification process, FeAl metallic ribbons are formed. These ribbons have a defined intermetallic structure, as already mentioned in previous works [17]. The ribbons are received in a container in the next module, the steam chamber, for the disintegration and conditioning of FeAl intermetallic powder.

### 2.1.3. Steam Chamber Modulus

The resulting intermetallic ribbons from the previous modulus are collected in the steam chamber to continue their physical treatment. The ribbons fall onto a container made of a stainless steel structure with a metallic mesh, as shown in Figure 1. Steam is injected through nozzles and fixed according to a configuration represented in Figure 1, in such a way that the contact between the steam and the metallic ribbons is maximized.

For this modulus, the steam is fed from a solar thermal energy system [21]. For solar industrial process heating (SIPH), there are three main types of collector technologies:

- Flat plate;
- Evacuated tube;
- Concentrators.

High-temperature collectors concentrate sunlight using mirrors or lenses and are generally used for fulfilling heat requirements up to 400 °C. The concentrators are of various designs, including linear Fresnel concentrator (LFC) [31], parabolic dish and parabolic trough. This concentrator is made up of strategically spaced mirrors so that the solar radiation that falls on the opening area is sent to an absorber located in the focal zone through which a working fluid circulates.

The industrial thermosolar system with a flat plate consists of several solar collectors with a double-tempered glass cover connected to each other, which heat the fluid and deliver it to the hot water tanks through forced circulation [21]. The industrial solar heater generates inside temperature ranges of 160 °C.

The steam chamber unit has also a collecting facility at its bottom to receive the intermetallic powder, which is screened and conveyed to the next modulus for conditioning. An important point to be observed is that the intermetallic powder does not agglomerate to form aggregates.

Special attention should be given to the possible electric static charges in the equipment, which will make it more difficult to keep the powder without aggregates.

2.1.4. Intermetallic Powder Receiving Modulus with Conditioning Chamber

Once the FeAl powder is obtained, it is conditioned in this last modulus. This unit delivers the product in its final state and specifications. It includes drying and sieving stages. The intermetallic powder is collected on a hopper and heated at a low temperature (30 °C) to eliminate moisture.

The described operations for this unit can be carried out with solar energy (T) for drying the powder and with solar-generated power (PV) for sieving operations.

## 3. Discussion

So-called "hybrid" cells are defined as PVT (photovoltaic thermal) systems that can generate both photovoltaic and thermal energy [32–35]. This system is the combination of a photovoltaic collector (PV) in the upper part and a thermal collector (T) in the bottom part. The different current technological trends are based on trying to increase the irradiation yield, decreasing heat and reflection losses and achieving a balance between thermal and photovoltaic efficiencies.

One major disadvantage of these systems is the fact that temperature rise has a decreasing effect on photovoltaic energy generation. This phenomenon makes it necessary to cool the PV system in a very efficient way, since the other part of the thermal collector system works continuously to heat water.

This system is still in the development and optimization stages [34]. Nevertheless, it is a system to be considered in the future to feed all process units of intermetallic powders, making this process a sustainable one, since besides being environmentally friendly, it saves energy. This solar system would help or supplement the energy for feeding the induction furnace, help the melt spinning equipment by supplying energy to keep the material molten and simultaneously help to rotate the cooling disk. It would supply the required steam for metallic ribbon disintegration and, finally, it would also supply the heat for drying the produced metallic powders.

In this case, a solar system that has the possibility of generating photovoltaic as well as thermal energies in the same module is highly advisable, because it is simpler to use and maintain and is more practical for the overall process architecture.

## 4. Conclusions

This communication proposes scaling-up the manufacturing process of an FeAl powder and of any other intermetallic with 55 to 60 at.% aluminum to produce up to 15 kg of intermetallic compound. The necessary equipment for the process scale-up consists mainly of:

(a)  Induction furnace of capacity up to 15 kg;
(b)  Rapid solidification equipment to produce metallic strips;
(c)  Steam chamber for strip disintegration to powder;
(d)  Powder drying chamber.

A layout proposal for this equipment is presented, and in order to make the process even more sustainable, solar energy supply in the different process stages is suggested as supplementary energy.

The conceptual design for the manufacture of intermetallic FeAl powders presented here includes an important contribution for sustainability through the incorporation of renewable energy application. This is an important factor for the future of this novel process to be taken into consideration for its scale-up stage and further development.

The option of compact process synthesis can be integrated step by step to help the economy of the process.

**Author Contributions:** Methodology, M.S.; validation, M.S. and F.-A.L.; formal analysis, F.-A.L.; investigation, M.S. and F.-A.L.; writing—original draft preparation, F.-A.L.; writing—review and editing, M.S. and F.-A.L.; visualization, M.S. and F.-A.L.; supervision, M.S.; project administration, M.S. All authors have read and agreed to the published version of the manuscript.

**Funding:** This research received no external funding.

**Institutional Review Board Statement:** Not applicable.

**Informed Consent Statement:** Not applicable.

**Data Availability Statement:** Not applicable.

**Conflicts of Interest:** The authors declare no conflict of interest. The funders had no role in the design of the study; in the collection, analyses, or interpretation of data; in the writing of the manuscript or in the decision to publish the results.

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
