# Peer review of "Conceptual Design of a Pilot Process for Manufacturing Aluminum-Based Intermetallic Compound Powders"

_2674-0516, doi:10.3390/powders2020030_

Round 1
Reviewer 1 Report
I have carefully read this paper entitled with “Conceptual Design of a Pilot Process for Manufacturing Aluminum-Based Intermetallic Compound". its interesting work, as a result, I have only a few minor points that the authors should address before it is accepted for publication. Please, publish subject to the following revisions:
1- Rewrite the novelty statement at the end of the introduction section.
2- Authors should justify the importance of the current work of how it is different from earlier reports. So, it’s better to add comparison table material and its performance to show the importance of the manuscript.
3- Abstract and conclusion should be rewritten and showed the clearer result of this study.
Reviewer 2 Report
The research content of this article is a design concept of FeAl alloy powder preparation process, which has certain value. I think there are still the following fatal problems.
1. In the introduction, the poor toughness of FeAl intermetallic compounds is not described in detail, nor does the design concept solves this problem.
2. How the FeAl ribbons was changed to powder in the team chamber unit is not stated.
3. The provision of energy with photovoltaic thermal makes people doubt whether the energy needs of the system can be achieved, especially since the device is used in the continuous production process, which is not explained and discussed in the paper.
4. The effect of the powder preparation was not discussed or proved by the data.
5. In the conclusion, the two conclusions are macroscopic, not the key to the design concept to solve powder preparation and improve their ductility.
Reviewer 3 Report
The short communication presents the Conceptual Design of a Pilot Process for Manufacturing Aluminum-Based Intermetallic Compound Powders. The concept is interesting. However, the following comments need to be considered before considering publication:
1. The authors mentioned that the technology is in TRL 3. Please justify.
2. What is the expected morphology and size of the powders produced through the route?
3. The authors need to explain the advantages of the present technique over other powder production techniques. A brief comparison is appreciated.
4. The authors need to comment on the purity control of the powders. How can the authors control the oxygen content?
5. Photographic view of the system can be added.
6. What is the typical steam velocity used in stage 3?
Round 2
Reviewer 2 Report
No other advice!
Reviewer 3 Report
The manuscript can be accepted.